# Genotypic and Phenotypic Characterization of *Pseudomonas atacamensis* EMP42 a PGPR Strain Obtained from the Rhizosphere of *Echinocactus platyacanthus* (Sweet Barrel)

**DOI:** 10.3390/microorganisms12081512

**Published:** 2024-07-24

**Authors:** Leilani Itzel Salinas-Virgen, María Eugenia de la Torre-Hernández, José Félix Aguirre-Garrido, Francisco Martínez-Abarca, Hugo César Ramírez-Saad

**Affiliations:** 1Doctorado en Ciencias Agropecuarias, Universidad Autónoma Metropolitana-Xochimilco, Mexico City 04960, Mexico; lisv.nani@gmail.com; 2CONAHCYT-Universidad Autónoma Metropolitana-Xochimilco, Mexico City 04960, Mexico; mdelatorre@correo.xoc.uam.mx; 3Departamento Sistemas Biológicos, Universidad Autónoma Metropolitana-Xochimilco, Mexico City 04960, Mexico; 4Departamento de Ciencias Ambientales, Universidad Autónoma Metropolitana-Lerma, Lerma de Villada 52004, Mexico; j.aguirre@correo.ler.uam.mx; 5Departamento de Microbiología del Suelo y la Planta, Estación Experimental del Zaidín, Consejo Superior de Investigaciones Científicas, 18008 Granada, Spain; fmabarca@eez.csic.es

**Keywords:** PGPR, *Pseudomonas atacamensis*, genomics

## Abstract

Plant growth-promoting rhizobacteria (PGPR) are a group of bacteria that associate with the rhizosphere of plants; one of the most abundant bacterial genera in this ecological niche is *Pseudomonas*, which is constantly expanding due to the emergence of new species such as *Pseudomonas atacamensis*, whose discovery in 2019 has led to the characterization of several strains from different environments but taxonomically related. The objective of this work was to phenotypically and molecularly characterize *P. atacamensis* strain EMP42, isolated from the rhizosphere of *Echinocactus platyacanthus*. The strain EMP42 is able to use different substrates and reduce oxidative stress in plants. It is capable of improving growth parameters such as the number of inflorescences and the height of the aerial body of *Arabidopsis thaliana*, as well as the germination and seedling survival of the cacti *Echinocactus platyacanthus* and *Astrophytum capricorne*. The genetic structure of *P. atacamensis* EMP42 consists of a closed chromosome of 6.14 Mbp, and 61.1% GC content. It has 5572 genes, including those associated with PGPR activities, such as the *trp*ABCDE, SAP, *pho*ABPRU and *acs*ABC genes, among others, and three ncRNA loci, nine regulatory regions, five complete rRNA operons and three CRISPR-Cas loci, showing phylogenomic similarities with the reference strain *P. atacamensis* B21-026. Therefore, this study contributes to the understanding of genomic diversity within *P. atacamensis* and, particularly, highlights the potential application of strain EMP42 as a PGPR.

## 1. Introduction

Plant growth-promoting rhizobacteria (PGPR) arouse great interest in the agroecological field since they are associated with plants’ rhizosphere, promoting their growth and development through various metabolic activities [1]. Using PGPR reduces the dependence on chemical fertilizers in crops of commercial interest, leading to sustainable agriculture [2].

One of the bacterial genera most reported as PGPR is *Pseudomonas* [3], which is also among the most abundant in the rhizosphere of plants [4]. Members of this genus exhibit a wide range of PGPR activities, including atmospheric nitrogen fixation, potassium solubilization, production of phosphatases and/or organic acids for phosphate solubilization, synthesis of phytohormones that enhance tissue development, production of siderophores for the uptake of metal ions, such as Fe, Ni, Cu, Zn, etc., as well as the production of antibiotics and various lytic enzymes that give them the ability to exert biological control over different phytopathogenic fungi [5].

The first report of *Pseudomonas atacamensis* dates back to 2020; this new species was isolated from the rhizosphere of flowering plants in the Atacama Desert, Chile [6]. Currently, 26 genomes belonging to strains within this species have been reported in GeneBank. They have been isolated from different environments, such as urban community gardens [7], clinical isolates [8], cow feces [9], agricultural soil [10] and plant-associated [11,12,13,14]. From these genomes, only six have been completely reconstructed. They belong to strains isolated from plants or soil, and some of them also have PGPR characteristics such as the ability to solubilize phosphates on strain SM1 [11] and to exert biological control on phytopathogenic microorganisms in strains B21-026, B21-045 and B21-050 [10].

The strain *Pseudomonas atacamensis* EMP42 was originally described as *Pseudomonas koreensis* strain P1 based on 16S rRNA gene sequencing of amplicons [15]. Due to its various in vitro PGPR features, this strain was further characterized in this work, including biochemical and in vivo PGPR features, sequencing of cloned 16S rRNA gene and a comparative genomic analysis.

## 2. Materials and Methods

### 2.1. Culturing and Phenotypic Characterization of the Bacteria

The current strain *P. atacamensis* EMP42 was isolated from the rhizosphere of *Echinocactus platyacanthus* (candy barrel), growing in the semidesert of Querétaro, Mexico [15]. It was cultured in TY liquid medium (tryptone 5 g/L, yeast extract 3 g/L, CaCl_2_ 1 g/L). The biochemical characterization of the bacteria was carried out with an API 20NE test kit, following the manufacturer’s instruction, adding oxidase and catalase tests.

In planta PGPR activity tests were carried out in 3 plant species in the model plant *Arabidopsis thaliana* and two cacti, *Echinocactus platyacanthus* and *Astrophytum capricorne*. In the former case, 3-week-old seedlings of wild-type *A. thaliana*, Columbia ecotype, were immersed (10 min) in a bacterial suspension of *P. atacamensis* EMP42 (OD_600nm_ = 0.6), while the group of non-inoculated seedlings was immersed in sterile-distilled water [16]. All seedlings were planted in plastic pots 6.5 × 6.5 × 8.5 cm (length × width × height), containing a previously tyndalized mixture of peat moss/agrolite/vermiculite (3:1:1). Pots were placed in a growth chamber for 16/8 h of light/dark, temperature 22 ± 3 °C, for 6 weeks. The following plant development parameters were recorded weekly: time of appearance of the main and secondary stems, time of appearance of inflorescences. The total number of stems, total number of inflorescences, survival percentage, diameter of the rosette and height of the aerial body were recorded at the end of the assay [16]. A combined *t*-test was performed to determine statistically significant differences between inoculated and non-inoculated plants.

The second plant assay was a germination test using seeds of the cacti *Echinocactus platyacanthus* and *Astrophytum capricorne*, collected in January 2021 and January 2024, respectively. The seeds (80 seeds from each species) were surface sterilized by immersion in 1% sodium hypochlorite for 15 min and rinsed several times with sterile water; then, they were divided into inoculated and non-inoculated groups. Seeds of the former groups were immersed (10 min) in a bacterial suspension of *P. atacamensis* EMP42 (OD_600nm_ = 0.6), while seeds of the non-inoculated groups were immersed in sterile-distilled water. Then, the 4 groups of seeds were sown in Petri dishes containing water-agar (1.2%) and incubated at 22 ± 3 °C. Four weeks after sowing, seed germination was recorded. The seedlings were transferred to a growth chamber for 4 more weeks for establishment and then transplanted into plastic trays containing commercial substrate for cacti (forest soil, coconut fiber and agrolite). Survival was evaluated 4 weeks later (12 weeks after the inoculation treatment). Differences in percentages of germination and survival of inoculated and non-inoculated seedlings were evaluated with an χ^2^ test.

### 2.2. Molecular Characterization

Genomic DNA extraction was carried out with the High Pure PCR Template Preparation Kit (Roche, Basel, Switzerland, catalog number 11796828001) following the manufacturer’s instructions. The 16S rRNA gene was amplified as previously described [15], and cleaned amplicons were cloned with the p-Gem T-easy kit (Promega, Madison, WI, USA, catalog number A1360), in competent *E. coli* DH5α cells under the manufacturer’s instructions. The transformed cells were selected in LB medium (tryptone 10 g/L, yeast extract 5 g/L, NaCl 10 g/L, agar 1.2%) supplemented with X-Gal (50 μg/mL), IPTG (0.1 mM) and ampicillin (100 μg/mL). After incubation at 37 °C for 24 h, transformant colonies were selected, and their respective plasmids were extracted by miniprep [17]. The plasmids were linearized with *Sca*I and their sizes evaluated by electrophoresis in 1.2% agarose gel. The cloned 16S rRNA genes were sent for Sanger sequencing at Macrogen, Korea. The complete gene sequences were uploaded to the 16S-based ID tool of the EzBioCloud web server [18]. A phylogenetic tree was constructed using the Maximum Likelihood method and the Jukes–Cantor as the best nucleotide substitution model [19], according to the jModelTest2 tool [20]. The resulting dendrogram was tested with a Bootstrap analysis of 1000 randomizations, all under the MEGA 11 software package [21].

Genomic DNA was sequenced on the Illumina MiSeq platform, at the Genome Sequencing Unit of CGEB-Integrated Microbiome Resource, Dalhousie University, Canada. A paired-end library was employed according to standard protocols [22]. The obtained reads (SRR29184006) were processed with Geneious Prime [23]. For quality filtering, reads with Phred score ≤ 20 and length < 150 nt were eliminated. The genome was reconstructed with the de novo assembly tool using the pre-established configuration, and the quality of the assembly was evaluated, both under the Geneious prime software (v 2023.2.1) [23]. Contigs > 1000 nt were mapped against the reference genome of *P. atacamensis* SM1 (CP070503.1), and gene order and orientation were determined with Mauve aligner [24]. The reconstructed genome was uploaded into the TrueBacID tool of EZBioCloud [25] for identification and deposited in the GeneBank under the accession number NZ_CP149965.

Finally, the annotation of the genes was performed with the NCBI-PGAP pipeline [26], the graphical representation of the genome was generated in Proksee [27], core-genome taxonomic analysis was performed on the OrthoVenn server [28], phylogenomic analysis was performed with FastTree2 [29] using the Maximum Likelihood method with the Jones–Taylor–Thorton (JTT) amino acid evolution model, with a CAT approximation of the rate of evolution for each site. The pangenome of *P. atacamensis* was determined with the Roary pipeline [30], by using only the complete genomes available of *P. atacamensis,* in order to avoid NP-hard [31]. The 7 genomes used for this analysis came from the following strains: SM1 [11], MGMM4 [32], SWIR7 [12], B21-026, B21-045 and B21-050 [10], and EMP42 (this work).

## 3. Results

### 3.1. Phenotypic Characterization

#### 3.1.1. Biochemical Characterization

The strain EMP42 was not capable of reducing nitrates to nitrites or nitrogen, fermenting glucose, hydrolyzing esculin (β-glucosidase activity), producing urease, β-galactosidase or assimilating maltose, adipic acid or acid phenylacetic. However, it showed the ability to assimilate different carbon sources, such as glucose, arabinose, mannose, mannitol, N-acetyl-glucosamine, potassium gluconate, capric acid, malic acid and citrate trisodium. In addition, it was able to form indoles, produce arginine dihydrolase, hydrolyze gelatin (protease activity) and also showed positive oxidase and catalase activities. Some in vitro PGPR capabilities were previously reported as *P. koreensis* P1.2 [4].

#### 3.1.2. In Planta Characterization

The ability of *P. atacamensis* strain EMP42 to promote plant development was assayed in seedlings of the fast-growing model plant *A. thaliana*. The results depicted in Table 1 show that inoculation with strain EMP42 has a beneficial effect on the development of the plant. Most of the quantitative parameters evaluated (i.e., appearance time of main stem and secondary stems, total number of stems, inflorescence appearance time, total number of inflorescences, rosette diameter and aerial body height) showed statistically significant differences, according to a combined *t*-test (*p* ≤ 0.05), while the categorical parameters. survival and rosette-forming plants (RF) did not produce significant differences, as tested by χ^2^.

In the inoculation assay of cacti seeds, statistically significant differences were found in most of the evaluated parameters, according to an χ^2^ test (Table 2). Inoculated seeds of both cacti species showed roughly a significant 15% increase in seed germination percentage, while inoculation of the 3-year-old *E. platyacanthus* seeds produced a two-fold increase in seedling survival, as compared to the non-inoculated groups.

### 3.2. 16S rRNA Gene Sequence Analysis

Sequencing of cloned 16S rRNA genes was meant to produce a more precise identification of strain EMP42. However, the four obtained sequences corresponded to two copies of the 16S rRNA genes, and a comparison with the EzBioCloud database showed that one copy of the 16S rRNA gene has greater similarity to the type strain *Pseudomonas atacamensis* M7D1^T^, and copy 2 with *Pseudomonas iranensis* SWRI54^T^ (Table 3). Furthermore, the phylogenetic relationships of the strain EMP42 and closely related *Pseudomonas* were evaluated with a phylogenetic tree based on 16S rRNA gene sequences (Figure 1). As expected, sequences of both copies of the EMP42 16S gene were grouped with the aforementioned type strains. These differences in identification pointed to the need for a deeper characterization based on genome sequencing.

### 3.3. Genome Assembly and Annotation

Whole-genome sequencing produced a total of 858,366 paired-end reads, with an average length of 246.7 bases, providing 99x genome coverage. The Phred score quality of 95.1% of the sequences was 30, and they were used to perform the de novo assembly, obtaining 110 good-quality contigs with an N50 length of 923,094 bp, an L50 of 2 contigs and a G+C content of 59.78%. The mapping of the contigs with length > 1000 bp culminated in the reconstruction of a closed genome of 6,145,328 bp size (Figure 2). With the complete genome sequence, a genome-based identification pointed out that strain EMP42 is indeed *P. atacamensis*, with an ANI of 96.82% according to the TruebackID tool [25]. Additionally, reconstruction of the EMP42 genome produced an extra-chromosomic contig with a different fold coverage, a sequence of 36,460 bp corresponding to the genome of a bacteriophage, with some other phage sequences were found scattered along the genome.

The genome annotation predicted 5572 genes, 3 ncRNA loci, 9 regulatory regions of cobalamin, guanidine, trehalose 6-phosphate phosphatases (TPP), S-adenosyl-L-homocystein, *yyb*P-*yko*Y for manganese sensing and flavin mononucleotide (FMN), plus 5 complete rRNA operons showing 99.94% similarity among their 16S sequences. Coding genes and operons associated with PGPR activities are depicted in Appendix A.

### 3.4. Comparative Genomics

The pangenome analysis of *P. atacamensis* was performed with the seven available complete genomes to avoid additional difficulties related to the use of fragmented assemblies [30] or scaffolding problems that may interfere with the generation of the pangenome [31]. Some genome features of the seven strains used for this analysis can be found in Table 4. *P. atacamensis* has an open and expanding pangenome (Appendix A) composed of 7712 genes, of which 3264 are part of the accessory genome. This latter is made up of 1104 genes shared by most of the genomes (*shell genome*), and 2160 genes shared by a minimal subset of the genomes (*cloud genome*), including those genes present in only one of the genomes (Figure 3A). The strain under study has a *core genome* consistent with the reference strains; interestingly, there is a specific metabolic signature in its cloud genome (Figure 3B), comprising genes for activities, such as the production of transposition proteins for genetic recombination, proteins related to DNA replication and metabolism and other proteins related to copper resistance and cobalt–zinc–cadmium resistance.

Grouping of all the genes in the *pangenome* generated 5504 protein-coding clusters. The strain EMP42 shares 5062, of which 4551 constitute the core genome, containing proteins with basic functions related to DNA synthesis, cell division and basic metabolic functions. The phylogenomic tree (Figure 4) showed that *P. atacamensis* strain EMP42 is phylogenomically closer to the strain *P. atacamensis* B21-026, as both strains have a similar number of orthogroups sharing most of them.

Figure 5 shows the exclusive or shared protein clusters present in the accessory genomes of six reference strains. Among the shared protein clusters are those related to PGPR functions, such as phosphatase production, tryptophan biosynthesis, xenobiotic metabolism, biofilm formation, siderophore transport, etc. *P. atacamensis* EMP42 also has seven exclusive clusters, with three related to DNA binding proteins, transport and response to cadmium stimulation, while the other four have unidentified biological functions.

## 4. Discussion

The results obtained from the biochemical characterization showed that *P. atacamensis* EMP42 has important characteristics as a PGPR, as it is able to assimilate nine different substrates. Bacterial plant growth promotion begins with the ability of bacteria to colonize the rhizosphere of plants, which is influenced by the type of exudates (i.e., sugars, amino acids and organic acids), serving as chemo-attractants and food for bacteria [33,34,35]. Furthermore, the strain shows the ability to produce metabolites and enzymes related to plant growth promotion activities, such as the production of indoles associated with the production of IAA; protease activity related to the biocontrol of phytopathogens, along with other lytic enzymes such as chitinases, cellulases and glucanases [36]; and the production of catalase that has an antioxidant function reducing oxidative stress in plants [37].

Plant growth-promoting characteristics among members of the *Pseudomonas* genus have been widely reported [5,38], agreeing with the previous evaluation of strain EMP42 regarding its ability to produce indole acetic acid, solubilize phosphates, produce siderophores for various metal ions, and exert biological control on phytopathogenic fungi [4]. In this context, comparison of strain EMP42 with strain *P. atacamensis* M7D1 showed that both share the aforementioned characteristics. However, strain EMP42 stands out for producing a 26% more indole acetic acid (IAA) than strain M7D1. Furthermore, the influence of various *Pseudomonas* species on the growth and development of different crops of commercial interest, such as canola [39], turmeric [40], mung beans [41] and corn [42], has been demonstrated. In model plants, such as *Brassica napa* [43] and *A. thaliana*, improving plants’ performance and increasing their biomass [44,45], these features were also observed in the present study. For this reason, members of the genus *Pseudomonas* have become the bioinoculant of choice in many countries worldwide [46].

The inoculation assay with *E. platyacanthus* and *A. capricorne* was focused on evaluating the germination of seeds and the seedling survival of these very slow-growing cacti (i.e., *E. platyacanthus* grows less than 5 cm per year [47]). In addition, *E. platyacanthus* is subject to special protection, and *A. capricorne* is a threatened species according to the Mexican normativity [48]; both species have relatively short germination times and high germination rates [49,50], which allow for timely and effective evaluation. The results show statistically significant increases of 15 and 12.5% in seed germination for E. *platyacanthus* and *A. capricorne*, respectively, as compared with their respective uninoculated controls. Different studies [51,52] have assayed several *Bacillus* spp., *Azospirillum brasilense*, or fungi (*Trichoderma* spp., *Glomus intraradices*), to stimulate the germination of some cacti seeds (*Mammillaria magnimamma*, *Pachycereus pringlei, E. platyacanthus*). Although their results varied according to the plant, bacteria or fungal species used, further differences regarding seed pretreatment or evaluation of germination time complicate making fair comparisons.

Seedling development could be influenced by the production of IAA by bacteria, since this auxin promotes cell division, as well as the elongation of the main root and the formation of secondary roots and root hairs [53]. This could also be related to the survival of plants, particularly for cacti, whose root system has an anchoring function in the rocky shallow soils they commonly inhabit [54]. In this study, a 47.8% increase in the survival of *E. platyacanthus* seedlings inoculated with the EMP42 strain was found, in contrast to the respective uninoculated control.

The molecular basis of the PGP capabilities of *P. atacamensis* EMP42 are related to the presence of genes encoding enzymes and proteins that are either directly or indirectly involved in these activities. Genes, such as *Fla*AB (Flagellin), *Flh*AB, *Fli*IR and *Flg*DH (synthesis of flagellar proteins) and *Che*AVY (chemotactic proteins), among others, intervene in the chemotactic response of bacteria and are essential for rhizosphere colonization [3].

In the genome of *P. atacamensis* EMP42, there are various genes involved in several biochemical pathways for IAA production [55], like the nitrilases N2 and N3 for the acetonitrile (IAN) pathway; the AO (monoamine oxidase), involved in the tryptamine (TAM) pathway; and IAD (indole-acetaldehyde dehydrogenase) needed in the indole-3-pyruvate (IPyA) pathway. These findings suggest that strain EMP42 might be capable of producing IAA through three different metabolic pathways, whose activation depends on the environmental stimulus [56]. It is worth mentioning that the IPyA pathway is mostly used by beneficial rhizobacteria [46].

We found other important genes that intervene in the solubilization and assimilation of phosphate from soil: SAP (alkaline phosphatase), IPP (inorganic pyrophosphatase), *pho*APRU (all make up the *pho* regulon whose products allow for detecting the availability of inorganic phosphate) [57] and *pts*ABCS (coding P transport proteins) [58]. The production of siderophores is represented by the presence of the *acs*ABCDE operon that encodes enzymes involved in the production of achromobactin, a siderophore of the citrate/carboxylate family first described in *Erwinia chrysanthemi* [59] and also found in *Pseudomonas* [60]. Additionally, according to PGAP annotation, EMP42 has a set of genes similar to *acs* that could participate in the production of a putative uncharacterized siderophore.

Furthermore, the presence in the cloud genome of strain EMP42 of genes involved in DNA transposition, plus an extrachromosomal bacteriophage sequence, is an interesting finding, since both types of elements may enhance the genetic variability of bacteria [61]. Among the 25 *P. atacamensis* genomes deposited in GeneBank, only that of strain SM1 contains a plasmid or extrachromosomal element, bearing sequences coding for bacteriophage structural proteins, such as stem and baseplate assembly proteins and terminases. However, the phage sequences found in the genomes of both strains do not have homology, either at the DNA level or at the protein level.

## 5. Conclusions

*P. atacamensis* strain EMP42 was isolated from the rhizosphere of candy barrel, and it showed both in vitro and in planta growth promotion characteristics, becoming an interesting strain with the potential to be used as a bioinoculant. A genomic comparison with genomes of other *P. atacamensis* strains obtained from rhizospheric or agricultural soils showed that the majority of the genes related to PGPR activities are found in the accessory genome and are not shared by all the strains studied; therefore, it is possible to establish the molecular bases of the differences on the PGPR capabilities of these bacteria.

It is important to further characterize the transposase and bacteriophage elements found in the genome of strain EMP42, in order to define their implications in rearrangements and/or transfer of genetic material, or even its fitness in the rhizosphere, as reported for *P. putida* by Quesada et al. [62].

Finally, given that the PGPR characteristics of *P. atacamensis* EMP42 were evaluated in planta under controlled conditions, the need arises to perform field tests that define its growth promotion capacity under real conditions, which would determine whether the strain could be used as a bioinoculant for crops different than cacti, including those of economic interest.

## Figures and Tables

**Figure 1 microorganisms-12-01512-f001:**
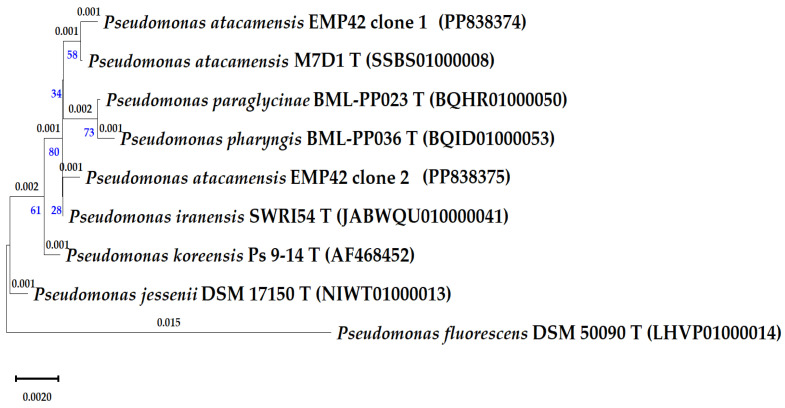
Phylogenetic tree of the *P. atacamensis* EMP42 and close *Pseudomonas*-type species, as based in partial 16S rRNA gene sequences (1250 pb). Phylogeny was inferred using the Maximum Likelihood method, the tree with the highest log likelihood (−3020.63) is shown. The percentage of trees in which the associated taxa clustered is shown below the branches in blue. The small branch lengths depicting the number of substitutions per site (values above branches) correspond to 1 or 2 variations in the gene sequences.

**Figure 2 microorganisms-12-01512-f002:**
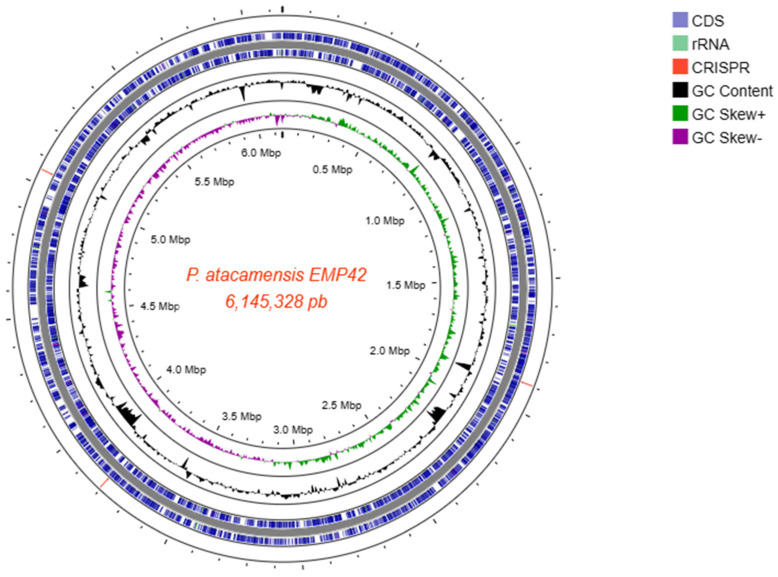
Closed genome map of *P. atacamensis* EMP42 with a length of 6,145,328 bp. From the outside: circle 1 showing the CRISPR-CAS loci (3 red lines); circles 2 and 3 enclose the genes on the main strand and complementary strand respectively; circle 4, the content of GC; circle 5, the GC bias. The scale in Mbp is indicated in the inner circle.

**Figure 3 microorganisms-12-01512-f003:**
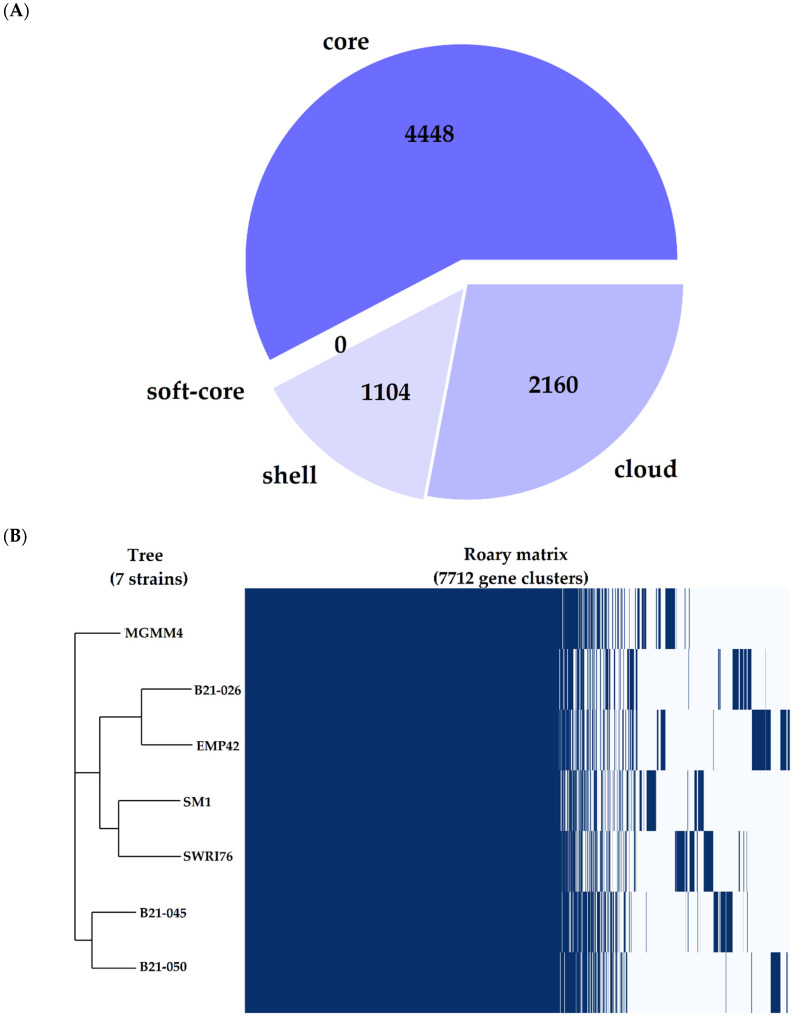
*Pangenome* representation of *Pseudomonas atacamensis*. (**A**) Pie chart representing the proportion of genes within the compared genomes. *Core* genes are found in >99% of the genomes, the *shell genome* is found in 15–95% and the *cloud genome* is represented in <15%. (**B**) Gene presence/absence matrix, each branch of the phylogenomic tree shows the genetic profile of each of the 7 strains compared.

**Figure 4 microorganisms-12-01512-f004:**
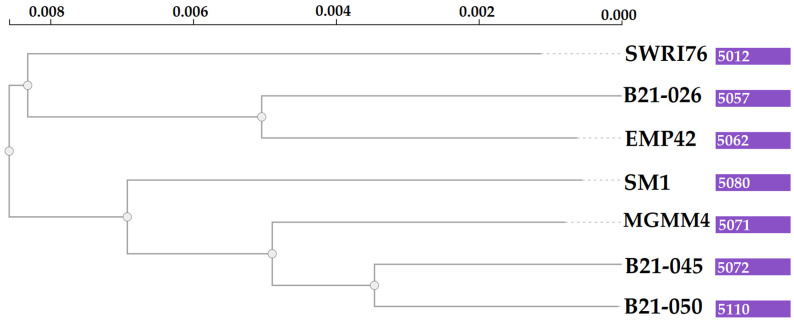
Phylogenomic tree of *P. atacamensis* strains. Values in the purple boxes are the orthogroup numbers of each analyzed strain.

**Figure 5 microorganisms-12-01512-f005:**
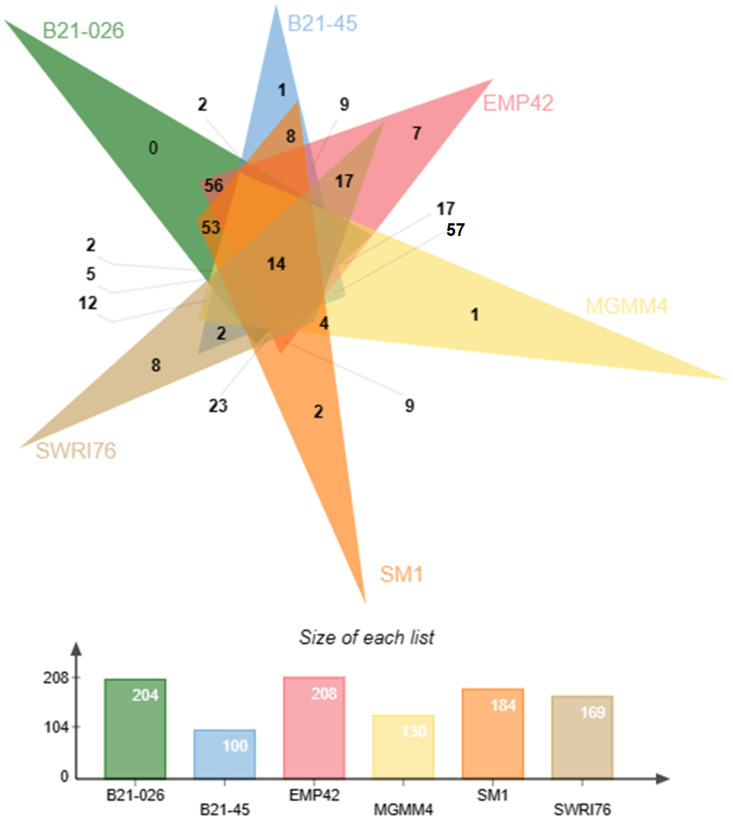
Venn diagram representation of the accesory genome of *P. atacamensis* strains EMP42, B21-026, B21-045, MGMM4, SM1 and SWRI76. Strain EMP42 (in pink) has 208 protein clusters most of them common to the other strains At the intersections of the triangles the numbers of shared clusters are shown, and the exclusive clusters for each strain are shown in the peaks. The number of clusters from each strain used to construct the Venn diagram are found in the lower bar graph.

**Table 1 microorganisms-12-01512-t001:** Development parameters of *A. thaliana* seedlings inoculated with *P. atacamensis* EMP42. The values correspond to the mean of 15 plants. The parameters with an * show significant differences.

Treatment	Survival (%)	RF (%)	MS * (Weeks)	SS * (Weeks)	IA (Weeks)	NS *	NI *	RD * (cm)	HAB * (cm)
Inoculated	100	100	2.57	4.83	2.57	6.85	36	2.59	13.46
Non-inoculated	91.67	90.91	3.27	5.43	3.27	2.72	15.6	2.23	10.68

Survival after 6 weeks; RF—plant-forming rosette; MS—mean of main stem appearance time; SS—mean of secondary stems appearance time; IA—mean of inflorescence appearance time; NS—mean number of stems; NI: mean number of inflorescences; RD—mean rosette diameter; HAB—mean of height of the aerial body.

**Table 2 microorganisms-12-01512-t002:** Development parameters in cacti seedlings of *E. platyacanthus* and *A. capricorne* inoculated with *P. atacamensis* EMP42. Parameters marked with * show statistically significant differences according a χ^2^ test.

Treatments	*E. platyacanthus*		*A. capricorne*	
	Germination * (%)	Survival * (%)	Germination * (%)	Survival (%)
Inoculated	95	92.86	100	81.81
Non-inoculated	80	45	87.5	83.87

**Table 3 microorganisms-12-01512-t003:** Sequence comparison of 2 cloned copies of the 16S rRNA gene of *P. atacamensis* EMP42 according to the 16S-based ID tool from EzBioCloud.

	Closest spp.	Accesion Number	Coverage (%)	Similarity (%)
Clone1 (PP838374)	*Pseudomonas atacamensis* M7D1^T^	SSBS01000008	100	99.92
Clone2 (PP838375)	*Pseudomonas iranensis* SWRI54^T^	JABWQU010000041	100	99.73

**Table 4 microorganisms-12-01512-t004:** Basic metrics of the 7 *Pseudomonas atacamensis* complete genomes used in this study. Rhizo strains were obtained from rhizospheric soil of different plants, while strains of series B21 were isolated from agricultural soil.

Strain	Isolation Source	GenBank Accesion	Length (Mbp)	GC (%)	CDS	Contigs	Coverage	Ref
*P. atacamensis* EMP42	Rhizo	CP149965.1	6.145	60	5572	1	99x	[15]
*P. atacamensis* SM1	Rhizo	CP070503.1	5.991	60	5469	2	200x	[11]
*P. atacamensis* MGMM4	Rhizo	CP123909.1	6.011	60	6011	1	150x	[9]
*P. atacamensis* SWIR7	Rhizo	CP077081.1	6.084	60	5461	1	121x	[12]
*P. atacamensis* B21-026	Soil	CP087187.1	5.970	60	5398	1	181x	[10]
*P. atacamensis* B21-045	Soil	CP087171.1	5.992	60	5405	1	139x	[10]
*P. atacamensis* B21-050	Soil	CP087167.1	5.940	60	5356	1	216x	[10]

## Data Availability

The genome sequence of *P. atacamensis* EMP42 is available in the GenBank database under accession number NZ_CP149965. The raw data are available on the same platform with the accession number SRR29184006. The sequences of cloned 16S rRNA gene have the accession numbers PP838374 and PP838375.

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
