# Peer review of "Genotypic and Phenotypic Characterization of Pseudomonas atacamensis EMP42 a PGPR Strain Obtained from the Rhizosphere of Echinocactus platyacanthus (Sweet Barrel)"

_microorganisms, 2024, doi:10.3390/microorganisms12081512_

Round 1

Reviewer 1 Report

Comments and Suggestions for Authors

An important group of microorganisms are plant growth-promoting rhizobacteria (PGPR). They have great potential to improve crop productivity and protect plants against biotic and abiotic stress. Chemicals used in agriculture are being replaced by biological control using beneficial microorganisms. These microorganisms have a beneficial effect on plants through various mechanisms of direct or indirect changes known to promote plant growth. One of the genera represented by PGPR is the genus Pseudomonas spp. Some Pseudomonas strains have properties in both plant growth promotion and phytopathogen control.

The article is based on the phenotypic and genomic analysis of the Pseudomons atacamensis EMP42 strain isolated from the rhizosphere of Echinocactus platyacanthus. The work is well written, although it contains a few errors.

Page 1 line 22 Please remove the dot after the word "related."

Page 2 line 47 Please remove the semicolon after “such as”.

Page 2 line 69 ''an API 20NE gallery'' It should be an API 20NE test kit

Page 2 lines 74, 90 It should be written ‘’OD’’, not “DO’’.

Page 2 I understand that the tested strain was isolated from the rhizosphere of Echinocactus platyacanthus and therefore the plant was used in in planta research. What was the criterion for choosing the second plant, Astrophytum capricorne?

Page 3 line 106 Please insert a semicolon in the name x-gal.

Page 3 line 10 In the name Sanger, we write ''r'' without italics.

Page 3 Please provide sequence and genome accession numbers.

Page 3 line 115 Please remove the dash.

Page 3 line 146 ''In Planta'' should be written in italics

Table 1 ''NS - mean number of stems (#); NI: mean number of inflorescences (X)'' – what do the symbols in brackets mean?

Page 4 lines 170, 172 Missing space in ''16SrRNA''

Page 4 line 177 There should be the full name of the gene.

Figure 1 The phylogenetic tree was created based on the full sequence of the 16S rRNA gene or a fragment of this gene. In the GenBank database, I found information that the deposited sequence has 1261 nucleotides.

Page 6 Please expand the abbreviation TPP; What is the regulatory region of manganese?

Table 4 "GenBank accesión" GenDoc is written without italics and please remove the accent

Page 10 line 288 Please remove the dot after the name of the bacteria

Comments on the Quality of English Language

-The language in the manuscript should be revised e.g.

punctuation (use of commas, semicolons)

syntax – line 143: there is no subject (what was able to form indoles?) (“In addition, was able to form indoles)

Grammar - l. 93: after 4 weeks seeds’ germination => after 4-week seed germination

Author Response

Comments 1: Page 1 line 22 Please remove the dot after the word "related."   

Responses 1: Done

Comments 2: Page 2 line 47 Please remove the semicolon after “such as”      

Responses 2: Done

Comments 3: Page 2 line 69 ''an API 20NE gallery'' It should be an API 20NE test kit  

Responses 3: Done

Comments 4: Page 2 lines 74, 90 It should be written ‘’OD’’, not “DO’’ 

Responses 4: Done

Comments 5: Page 2 I understand that the tested strain was isolated from the rhizosphere of Echinocactus platyacanthus and therefore the plant was used in in planta research. What was the criterion for choosing the second plant, Astrophytum capricorne?    

Responses 5: Information in this regard was added in the Discussion section lines 290-293

Comments 6: Page 3 line 106 Please insert a semicolon in the name x-gal       

Responses 6: Done

Comments 7: Page 3 line 10 In the name Sanger, we write ''r'' without italics  

Responses 7: Done

Comments 8: Page 3 Please provide sequence and genome accession numbers       

Responses 8: Done

Comments 9: Page 3 line 115 Please remove the dash

Responses 9: Done

Comments 10: Page 3 line 146 ''In Planta'' should be written in italics    

Responses: Done, this modification is now on line 147

Comments 11: Table 1 ''NS - mean number of stems (#); NI: mean number of inflorescences (X)'' – what do the symbols in brackets mean?    

Responses 11: The symbols were eliminated and several corrections introduced in the heading and footnote of Table 1. We hope now it is clearer

Comments 12: Page 4 lines 170, 172 Missing space in ''16SrRNA''             

Responses 12: Done, these modifications are now on lines 172 and 174

Comments 13: Page 4 line 177 There should be the full name of the gene            

Responses 13: Done, this modification is now on line 175

Comments 14: Figure 1 The phylogenetic tree was created based on the full sequence of the 16S rRNA gene or a fragment of this gene. In the GenBank database, I found information that the deposited sequence has 1261 nucleotides       

Responses 14: We thank the reviewer for this observation. The correct information has been added to the footnote of Fig 1

Comments 15: Page 6 Please expand the abbreviation TPP; What is the regulatory region of manganese?

Responses 15: The required information has been added on lines 209-210

Comments 16: Table 4 "GenBank accesión" GenDoc is written without italics and please remove the accent

Responses 16: Done

Comments 17: Page 10 line 288 Please remove the dot after the name of the bacteria              

Responses 17: Done. This modification is now on line 297

Comments on the Quality of English Language

-The language in the manuscript should be revised e.g. punctuation (use of commas, semicolons)

syntax – line 143: there is no subject (what was able to form indoles?) (“In addition, was able to form indoles).

Grammar - l. 93: after 4 weeks seeds’ germination => after 4-week seed germination

These corrections were done, and in order to improve the use of English language, some other modifications were introduced along the paper (all them highlighted), also several

Reviewer 2 Report

Comments and Suggestions for Authors

Interesting results are presented in the context of modern trends in whole-genome bacterial identification, their beneficial properties, and interactions with plants (undoubtedly, strain EMP24 can be classified as PGPR microorganism). These data, referenced by the authors, should be supported by in vitro results and classical methods.

However, it is not clear from the reference to source [15] which strains of Pseudomonas koreensis are being discussed (there are two of them, P1 and P18), as originally described strain P. atacamensis EMP42.

The same applies to the reference to source [4]. There, the PGP properties of three strains of P. koreensis (P1.1, P1.2, and P18.2) with differing properties are mentioned.

After correctly correlating the references to specific strains, I recommend the article for publication.

Author Response

Comments 1: It is not clear from the reference to source [15] which strains of Pseudomonas koreensis are being discussed (there are two of them, P1 and P18), as originally described strain P. atacamensis EMP42. The same applies to the reference to source [4]. There, the PGP properties of three strains of P. koreensis (P1.1, P1.2, and P18.2) with differing properties are mentioned       

Response 1: The original ID of Pseudomonas atacamensis EMP42 in two references is P1.2. This information was added on lines 60 and 146